# Multi-Destination Path Planning Method Research of Mobile Robots Based on Goal of Passing through the Fewest Obstacles

Hongchao Zhuang [1,*], Kailun Dong [1,*], Yuming Qi [1], Ning Wang [2] and Lei Dong [1]

1    Institute of Robotics and Intelligent Equipment, School of Mechanical Engineering,
     Tianjin University of Technology and Education, Tianjin 300222, China; chigym@163.com (Y.Q.);
     dongleihit@126.com (L.D.)
2    School of Information Technology Engineering, Tianjin University of Technology and Education,
     Tianjin 300222, China; wangning811108@163.com
*    Correspondence: zhuanghongchao_hit@163.com (H.Z.); dongkailun4869@163.com (K.D.);
     Tel.: +86-022-88181083 (H.Z.)

**Abstract:** In order to effectively solve the inefficient path planning problem of mobile robots traveling in multiple destinations, a multi-destination global path planning algorithm is proposed based on the optimal obstacle value. A grid map is built to simulate the real working environment of mobile robots. Based on the rules of the live chess game in Go, the grid map is optimized and reconstructed. This grid of environment and the obstacle values of grid environment between each two destination points are obtained. Using the simulated annealing strategy, the optimization of multi-destination arrival sequence for the mobile robot is implemented by combining with the obstacle value between two destination points. The optimal mobile node of path planning is gained. According to the Q-learning algorithm, the parameters of the reward function are optimized to obtain the q value of the path. The optimal path of multiple destinations is acquired when mobile robots can pass through the fewest obstacles. The multi-destination path planning simulation of the mobile robot is implemented by MATLAB software (Natick, MA, USA, R2016b) under multiple working conditions. The Pareto numerical graph is obtained. According to comparing multi-destination global planning with single-destination path planning under the multiple working conditions, the length of path in multi-destination global planning is reduced by 22% compared with the average length of the single-destination path planning algorithm. The results show that the multi-destination global path planning method of the mobile robot based on the optimal obstacle value is reasonable and effective. Multi-destination path planning method proposed in this article is conducive to improve the terrain adaptability of mobile robots.

**Keywords:** mobile robot; multi-destination path planning; obstacle value; simulated annealing strategy; Q-learning algorithm

## 1. Introduction

The innovation and optimization of artificial intelligence technology promotes the gradual development of mobile robots in the direction of automation and intelligence [1]. At the same time, with the demand of humans, mobile robots will move from the laboratory environment to outdoor environment. In order to ensure that mobile robots have high terrain adaptability in outdoor environments, the research on path planning of mobile robots has gradually attracted the interest of many scholars. The main purpose of mobile robot path planning is to select an optimal or sub-optimal collision-free path from the starting point to the end point for the mobile robot to move by exploring its environment. The core content of its research is the design and optimization of the planning algorithm. Path planning algorithms determine the performance index function according to the mobile requirements of the robot. They can plan a collision-free travel route [2]. Traditional path planning methods for mobile robots include artificial potential field method [3], fuzzy

logic algorithm [4], genetic algorithm [5], particle swarm optimization algorithm [6] and so on. The traditional path planning algorithm is mainly single-destination path planning. The multi-destination path planning algorithm is the integration and improvement of the traditional path planning. The multi-destination path planning for mobile robots is defined by Marek Cuchý. It is a collision-free and feasible path for mobile robots to move from a starting point to two or more destinations [7–12]. Although the traditional path planning method can also be used for multi-destination path planning, there are problems. For example, the path from the starting point to each destination needs to be planned many times. The problem of redundant path and even invalid path results in low efficiency of mobile robots and waste of energy. Therefore, in a multi-destination environment, obtaining a complete path that can reach each destination one by one has great significance for the mobile robot to save running time. At same time, it also improves planning efficiency and energy utilization.

According to the external environment information obtained by mobile robots, the path can be divided into two types: local and global. Global path planning is to plan a path for a robot in a known environment. The accuracy of path planning depends on the accuracy of the environmental information acquired. Global path planning can find the optimal solution, but it needs to know the accurate information of the global environment in advance. If the environment is not constructed accurately, it will affect the accuracy of global path planning. This problem even threatens the safety of the robot and the surrounding environment.

The multi-destination global path planning problem mainly involves quickly sorting the destinations, to quickly search the path and to ensure that the path planning has good convergence. This previous method was multi-destination transportation route planning. It used Hamiltonian chart construction conditions to construct a Hamiltonian path. This method obtained a path that traverses all destinations without repetition to achieve the optimal effect of multi-destination route planning [13]. The particle swarm optimization path planning method combined the fast convergence characteristics of the ant colony algorithm. This method converted the choice of target location into a traveling salesman problem and optimized it by ant colony algorithm [14]. The traversal multi-destination path planning method combined particle swarm, genetic and A-star algorithm. It solved the problem that the computational cost of the particle swarm-genetic algorithm was too high, and the mobile robot did not avoid obstacles when a single algorithm traverses multi-task target points [15]. The multi-destination path planning algorithm combined global static and local dynamics. It obtained an optimal path planning that enabled mobile robots to stabilize, avoided obstacles in time, and accurately moved to the destination [16]. The multi-objective intelligent water drop algorithm (MO-IWD) was based on the coefficient of variation (CV). This method had good diversity at the same time [17]. The non-dominated sorting genetic algorithm II (NSGA-II) enabled mobile robots to determine the best route for multi-objective problems with minimal cost [18].

The above-mentioned multi-destination global route planning research has verified that multi-destination route planning was more efficient than multiple single-destination route planning. However, the multi-destination path planning algorithm is more likely to fall into the local optimum. In the global path planning, these problems are low utilization rate of environmental information, complicated calculation structure, low sorting efficiency, large amount of calculation, and slow operation speed.

In view of the above problems, this article takes wheeled mobile robots as an example to propose the global multi-destination path planning method. The global multi-destination path planning method is based on the goal of passing through the fewest obstacles. It includes the reconstruction of environmental maps based on the rules of the Go chess game, the sorting of simulated annealing strategy and the path planning of Q-learning optimization algorithm. Its structure is shown in Figure 1. The sections of this article are arranged as follows. In the second part, the grid map is built to simulate the real environment. Based on the rules of go, the grid map is optimized and reconstructed to

obtain the grid environment and obstacle values between the two destinations. The obstacle values between the two destinations are used to optimize the arrival sequence of multi-destination robots by the simulated annealing strategy. The optimal mobile node for robot path planning is obtained. In the third part, the article optimizes the Q-learning algorithm to obtain the mathematical model of the mobile robot path planning. The mathematical model can obtain the optimal path. The optimal path for multiple destinations has the fewest obstacles. In the fourth part, the multi-destination path planning simulation of mobile robots is implemented by using MATLAB software to verify the rationality and effectiveness of the proposed method.

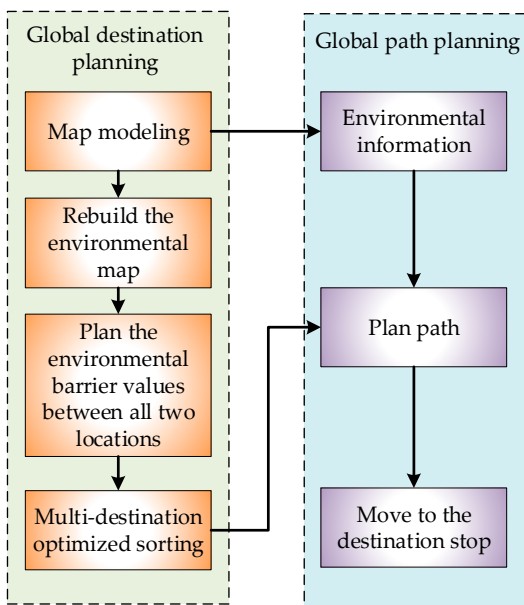

**Figure 1.** Algorithm structure of multi-destination path planning.

## 2. Global Destination Arrival Sequence Planning

For multi-destination path planning of mobile robots, it is necessary not only to plan the path between two destinations, but also to consider the order of arrival of each destination, consider the global shortest path and time efficiency. The constructed multi-destination path planning scheme is shown in Figure 2. In Figure 2, $S$ is the starting point. $T_i$ ($i$ = 1, 2, 3, 4) represents multiple destination points. Plan 1 and Plan 2 are two plans for multiple global multi-destination sequential arrival sequence planning. According to the global optimal path, the mobile robot can move.

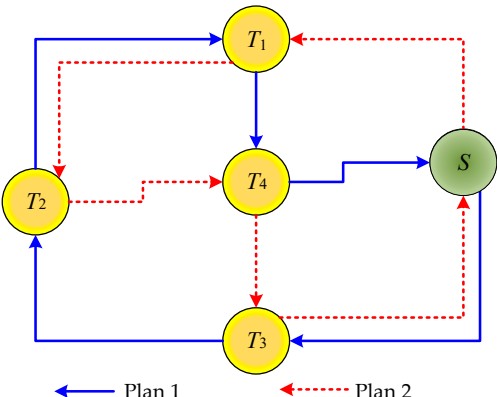

**Figure 2.** Multi-destination path planning schemes.

Global destination arrival sequence planning refers to the global planning of the arrival sequence of the robot from the starting point to all destinations before moving. First, it is necessary to calculate the environmental obstacle values between each starting location and each destination. Then, according to the calculated environmental obstacle value between the two locations, this article optimizes the arrival sequence of multiple destinations. In the end, the global path was chosen. As shown in Figure 2, it has the smallest environmental obstacle value. Global destination arrival sequence planning reduces the amount of calculation. The planned global path passes through all destinations at once. The obstacles of passing through are reduced. The length of the path is greatly reduced.

As shown in Figure 2, such as $S \rightarrow T_1 \rightarrow T_2 \rightarrow T_4 \rightarrow T_3 \rightarrow S$, $S \rightarrow T_3 \rightarrow T_2 \rightarrow T_1 \rightarrow T_4 \rightarrow S$, ..., there are many situations in the movement sequence. Therefore, it is necessary to optimize the ordering of multiple destinations. The selected global path can pass through all destinations with the fewest obstacles. It has a short distance. Therefore, this article first adopts the rules of the Go chess game to reconstruct the environmental grid map, as shown in Figure 3. Secondly, the matrix of the reconstructed environmental grid map is used to obtain the obstacle value of the environment between all two locations. The obstacle values are sorted, and those with the same environmental obstacle value can be processed twice. Assuming that there are $n$ destinations in the multi-destination path planning of a mobile robot, there are $(n^2 + n)/2$ combinations of the path arrival sequence, and there are also $(n^2 + n)/2$ environmental obstacle values. Aiming at meeting the shortest motion path requirements of the mobile robot, the robot needs to select an optimal path from $(n^2 + n)/2$ combinations. This article follows the environmental obstacle value to choose the optimal path.

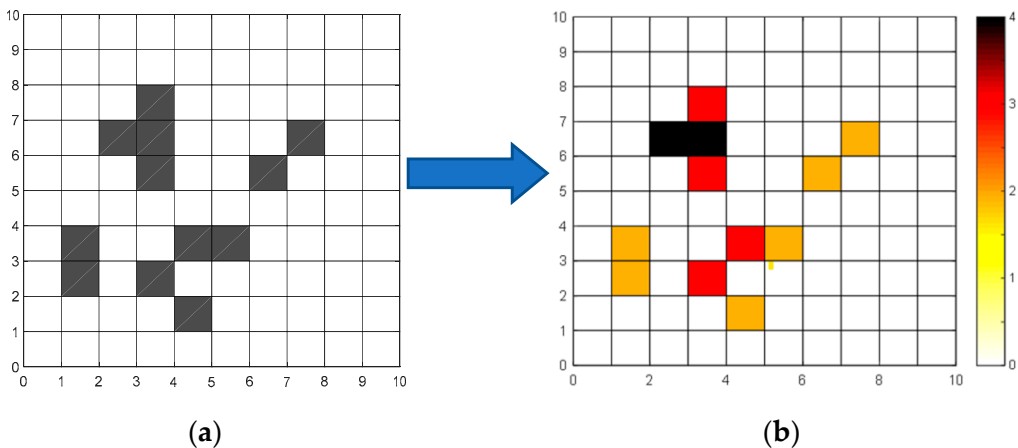

(**a**) (**b**)

**Figure 3.** Reconstruction of environmental grid map. (**a**) Original environment raster map. (**b**) Environmental grid map with obstacle values.

As shown in Figure 4, global destination planning can be divided into several steps and the several concrete steps are shown in the following Sections 2.1 and 2.2.

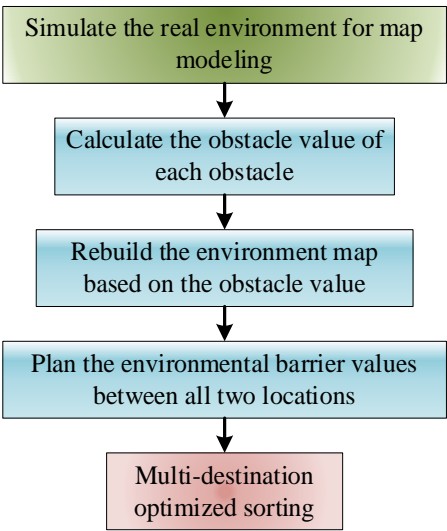

**Figure 4.** Flow of global destination planning.

## 2.1. Grid Map Optimization Based on the Rules of the Chess Lively Game of Go

Step 1. In order to make the experimental environment matrix $M$ without loss of generality, it is built in three parts. The first part simulates regularly arranged obstacles. The second part simulates a simple obstacle environment. The third part simulates a complex obstacle environment. This kind of environment is a metric grid as shown in Figure 5.

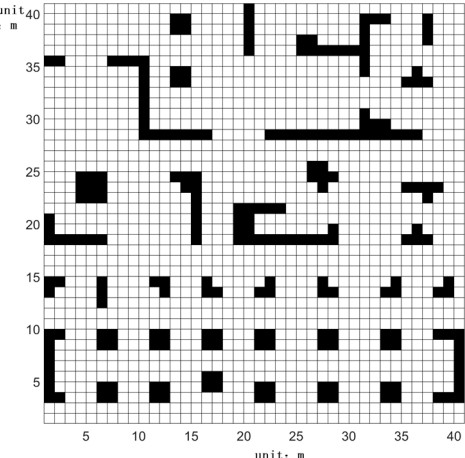

**Figure 5.** Specific environmental grid map of simulating the real environment.

Definition: Adjacent grids refer to the grid in front, back, left, right, left front, right front, left rear and right rear. The values of the grid map matrix are used to represent environmental obstacle values.

Step 2. Reconstructing the environment grid map determines the obstacle value of each grid. The grid's environmental obstacle value is calculated, according to Equation (1). If the grid's environmental obstacle value is not 0, it becomes the sum of this grid and adjacent grids. If the grid's environmental obstacle value is 0, it is unchanged. According to Equation (1), the environment map matrix $Me^{40\times40}$ with environmental obstacle values is constructed.

$$Me_{i,j} = \begin{cases} 0 & (M_{i,j} = 0) \\ M_{i-1,j-1} + M_{i-1,j} + M_{i-1,j+1} + M_{i,j-1} + M_{i,j} + M_{i,j+1} + M_{i+1,j-1} + M_{i+1,j} + M_{i+1,j+1} & (M_{i,j} = 1) \end{cases}, \quad (1)$$

where $M$ is the grid's environmental matrix. $M_{i,j}$ is the element of row $i$ and column $j$ of the $M$. The range of $i$ and $j$ is from 1 to 40. $M_{i,j}$ is 0, when $i > 40, i \le 0, j \le 0$ or $j > 40$.

Step 3. The coordinates of two points are $(m_1, n_1)$ and $(m_2, n_2)$ ($m$ is the number of rows and $n$ is the number of columns). The environment matrix $Me(m_1 : m_2, n_1 : n_2)$ between the two points is extracted in turn. According to Equation (2) the environmental obstacle value between each point is calculated.

$$G_i = \frac{sum(Me(\ m_1 : m_2 \quad n_1 : n_2\ ))}{|m_1 - m_2| \times |n_1 - n_2|}, \tag{2}$$

where $G_i$ is the $i$-th environmental obstacle value. $Me(\ m_1 : m_2 \quad n_1 : n_2\ )$ is the submatrix of the matrix $Me$ from row $m_1$ to row $m_2$ and from column $n_1$ to column $n_2$.

Step 4. The environmental obstacle value between each destination point $G_i$ ($i$ = 1, 2, ..., $n$) is obtained.

*2.2. Multi-Destination Ranking Method Based on Environmental Obstacle Value*

Definition: Robustness means that the results should not be greatly affected by data disturbances, noise and outliers present in the algorithm.

The simulated annealing algorithm [19] has few parameters, short calculation time, and robustness. It is suitable for the high efficiency requirements of this article. The multi-destination sorting process adopts the simulated annealing algorithm to optimize the sorting. Steps of the simulated annealing multi-destination sorting algorithm based on the environmental obstacle value are as follows.

Step 1. The initial temperature $Te_{max}$, the end temperature $Te_{min}$ and the cooling rate $r = 0.8$ ($0 < r < 1$) are determined as Equation (3). The starting point and $n$ destination points form the matrix $(S, T_1, T_2, ..., T_n, S)$.

$$Te_{max} = \sqrt{\sum_{i=1}^{n} G_i^2} \qquad Te_{max} - Te_{min} = n \times Te_{max}(1 - r) \qquad , \tag{3}$$

where $G_i$ is the $i$-th environmental obstacle value.

Step 2. The elements in the matrix $(S, T_1, T_2, ..., T_n, S)$ are rearranged to form the solution spaces $I^{n! \times (n+2)}$ as shown in the Equation (4). The number of solutions in the solution spaces $I^{n! \times (n+2)}$ is $n!$.

$$I = \begin{pmatrix} S & T_1 & T_2 & \cdots & T_n & S \\ \vdots & \vdots & \vdots & \ddots & \vdots & \vdots \\ S & T_2 & T_1 & \cdots & T_n & S \\ S & T_n & T_1 & \cdots & T_{n-1} & S \end{pmatrix}, \tag{4}$$

Additionally, the solution space $I$ randomly generates an initial solution $I_0$ ($I_0 \in I$).

Step 3. Generate a new solution through the two-transformation method. The selected sequence number is m, k. The access sequence between the $m$-th and $k$-th solution elements is exchanged. If the previous path solution of the exchange is $I_0 = (S, T_1, T_2, ..., T_m, ..., T_k, ..., T_n, S)$, the exchanged path is a new path, as shown in Equation (5).

$$I_i' = (\quad S \quad T_1 \quad T_2 \quad \cdots \quad T_k \quad \cdots \quad T_m \quad \cdots \quad T_n \quad S\ ), \tag{5}$$

Step 4. This step sets the environmental obstacle value as the objective function. The objective function is $f(S, T_1, T_2, ..., T_n, S)$. The difference between the objective function of solution $I_0$ before transformation and that of solution $I_i'$ after transformation is calculated from Equations (6) and (7).

$$f(S, T_1, T_2, ..., T_n, S) = \sum_{i=1}^{n+2} d(c_i, c_{i+1}), \tag{6}$$

$$\Delta f = f(I_i') - f(I_i), \tag{7}$$

where the difference between the objective function of the solution before transformation and that of solution after transformation is $\Delta f$. $c_i$ and $c_{i+1}$ are two adjacent target points, respectively. $d(c_i, c_{i+1})$ is the obstacle value of two adjacent target points.

Definition: accepting a new state with probability instead of using completely determined rules is called Metropolis criterion.

Step 5. According to Equation (8), the acceptance probability $p$ is calculated. The Equation (8) accepts the rule of Metropolis criterion to select the next generation of new solutions. If $\Delta f < 0$, it means that the cost of the solution after transformation is less than the solution before transformation, so the new solution is $I_i'$. If $\Delta f > 0$, it means that the cost of the solution after transformation is greater than the solution before transformation. The new solution is judged by accepting the current path as the new path probability $p$. The greater the probability $p$, the greater the possibility of acceptance.

$$p = \begin{cases} 1 & (\Delta f < 0) \\ \exp(-\Delta f / Te) & (\Delta f > 0) \end{cases}, \tag{8}$$

This step has a certain probability to accept the solution that is worse than the current solution, so it can jump out of the local optimum to a certain extent. The solution obtained by the algorithm is independent of the initial solution $I_0$ ($I_0 \in I$), so it has certain robustness.

Step 6. Equation (9) is used to update the temperature $Te$ by cooling the temperature down.

$$Te' = Te \times r, \tag{9}$$

where the temperature before updating is $Te$. The temperature after updating is $Te'$. $r$ ($0 < r < 1$) is the cooling rate.

Step 7. This step determines whether the temperature $Te'$ after cooling down has reached the end temperature $Te_{min}$. If the conditions are met, the optimal solution will be output and the algorithm will end. Otherwise, it goes to step 5.

This article uses the simulated annealing method to obtain the optimal solution. The optimal solution is the order of arrival of global multi-destination with the fewest obstacles. This sequence arrangement is ready for the next step of global multi-destination path planning. The simulated annealing method greatly improves the efficiency.

## 3. Multiple Destination Global Path Planning Strategy Based on Improved Q-Learning Algorithm

### 3.1. Q-Learning Algorithm Optimization

The traditional path planning algorithm needs to perform adjacency transformation on the environment grid map matrix to obtain the adjacency transformation matrix. It mainly represents the feasibility of the agent moving to each grid and ensures the convergence. For the method of sequence arrangements designed in this article, the environmental grid map matrix was reconstructed. It increases the amount of map information of the environmental model. The traditional algorithm also repeats the analysis of the environmental model. It is too redundant and cannot guarantee the convergence. The path planning of the Q-learning algorithm has nothing to do with the initial value. The Q-learning algorithm can guarantee the convergence without the environmental model. Therefore, this article uses the Q-learning algorithm for path planning.

Q-learning is a type of reinforcement learning algorithm. Reinforcement learning is mainly composed of agent, environment, state, action, and reward value [20]. It is a learning process to achieve a goal through multi-step decision making. The basic structure of reinforcement learning is shown in Figure 6. The Q-learning algorithm is based on the reinforcement learning framework. First, a Q value table is set, and the initial state and reward value are defined. After the agent performs an action, the environment will transform to a new state. In the new state, the environment will give the corresponding

reward value. Then, the agent calculates and updates the Q value table according to the new state and the reward of the environment feedback. According to the Q value table, the strategy is selected, and the new action is executed. Finally, through continuous interaction with the environment, the optimal action set is found. The flow of the Q-learning algorithm is shown in Figure 7. The simulation effect is shown in Figure 8.

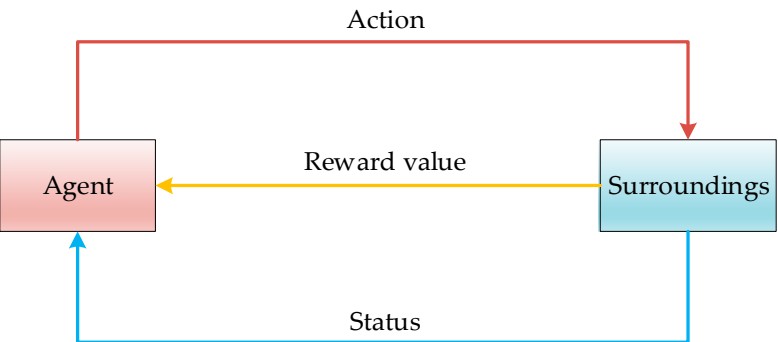

**Figure 6.** Basic architecture of reinforcement learning.

Initialization For any $s \in S$
Repeat Experience trajectory;
　　Initialization state;
　　Repeat Time step in experience trajectory;
　　　　According to the action state value $q$, Select action $a$ in state $s$
　　　　Perform action $a$, get reward $r$ and next time step state $s'$
　　　　$q(s,a) \leftarrow (1-\alpha)q(s,a) + \alpha[r + \gamma \, max_a \, q \, (s',a')]$, Update action value function
　　　　$s \leftarrow s'$, Record status
　　Until End time step $s$;
Output Action value function $q(s,a)$

**Figure 7.** Q-learning algorithm flow chart.

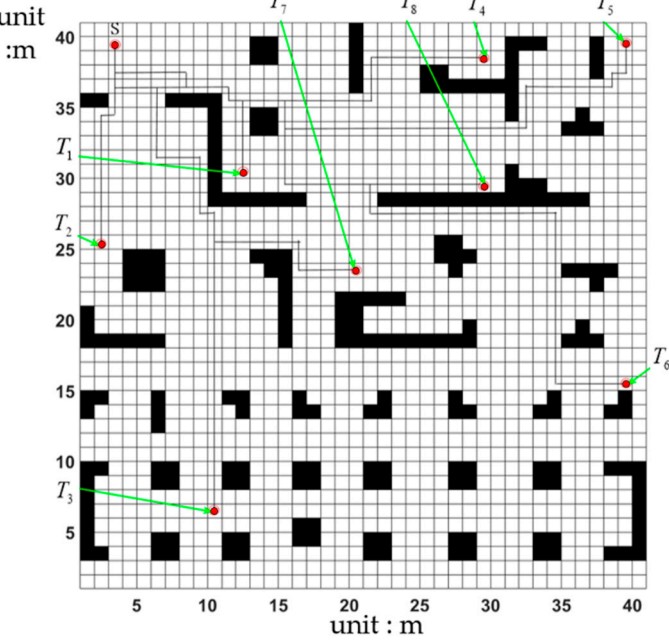

**Figure 8.** Q-learning algorithm effect diagram.

The following are the optimization measures in the algorithm.

It can be seen from the algorithm flow in Figure 7 that Q-learning uses the action value function to update the Q value table. The Q value table can be used to select the optimal strategy. However, Q-learning is essentially a greedy algorithm. There is a behavior that takes the highest expected reward every time. During the training process, it cannot explore other possible behaviors. The algorithm enters the "local optimum" and fails to complete the path planning. Therefore, this article set the coefficient ***path*** and the safety distance ***D*** as shown in Equation (10). In this way, the agent has a certain probability to take the optimal behavior. The efficient ***path*** stores the actions. The algorithm also has a certain probability to take all the actions that can be taken immediately. The traversed path is incorporated into the memory bank to avoid local optima and small-scale loops.

$$
\begin{cases}
r(s') = -1 & s' \in path \quad or \quad distence > D \\
r(s') = 0 & s' \notin path \quad or \quad distence \leq D
\end{cases}
, \tag{10}
$$

where $s'$ is the next state. $r(s')$ is the reward of the next state $s'$. The coefficient is ***path*** and the safety distance is ***D.*** *distence* is distance from obstacles.

In order to increase diagonal movement, this article sets the reward value of diagonal movement to take an approximate value of 0.707 as shown in Equation (11). It avoids the situation where the robot first moves to the upper left and then to the lower left instead of directly moving two squares to the left. Setting this value is determined based on the relative distance between two grids on the map.

Definition: The right downward motion is $(i, i + n + 1)$. The left downward motion is $(i, i + n - 1)$. The right upward motion is $(i, i - n + 1)$. The left upward motion is $(i, i - n - 1)$. $i = 1, 2, \ldots 40$.

$$
\begin{cases}
r(i, i + n + 1) = 0.707 & i + n + 1 \leq 40 \quad and \quad M(i + n + 1) = 0 \\
r(i, i + n - 1) = 0.707 & i + n - 1 > 0 \quad and \quad M(i + n - 1) = 0 \\
r(i, i - n + 1) = 0.707 & i - n + 1 > 0 \quad and \quad M(i - n + 1) = 0 \\
r(i, i - n - 1) = 0.707 & i - n - 1 > 0 \quad and \quad M(i - n - 1) = 0
\end{cases}
, \tag{11}
$$

### 3.2. Path Planning Based on Improved Q-Learning Algorithm

Global path planning means that the robot plans a global path from the starting point through all destinations while moving. The Q-learning algorithm has little connection with the initial value. It can guarantee the convergence without the environment model. This feature can be applied to the path planning of the mobile robot in this article. The Q-learning algorithm is based on the reinforcement learning framework. Firstly, a Q value table is set. The rows of the table represent different states. The columns of it represent different actions that can be taken. The initial state and reward value are defined, and the agent is executed. After a certain action, the environment will transform to a new state. For this new state, the environment will give a corresponding reward value. Then, the agent calculates and updates the Q value table according to the new state and reward feedback from the environment. The Q value table selects strategies and executes new actions. Finally, an optimal action set is found through continuous interaction with the environment.

Aiming at the characteristics of multi-destination path planning and the possibility of wheeled mobile robots, the overall destination arrival sequence planning is obtained. The improved Q-learning algorithm is used to complete the global path planning. The structure is shown in Figure 9.

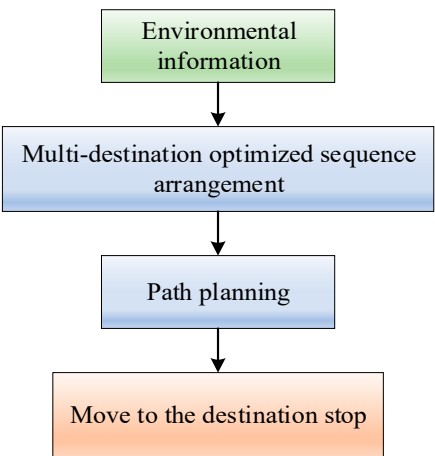

**Figure 9.** The flow of global path planning.

As shown in Figure 9, global path planning can be divided into several steps. The several concrete steps are shown as following.

Step 1. Initialize the Q value table. The Q value table has $n$ rows and $n$ columns ($n$ is the number of destinations). All values in the table are initialized to zero. The Q-value table is updated by Equation (12). Therefore, the basic form of the algorithm is Equation (12).

$$q(s,a) = q(s,a) + \alpha[r + \gamma max_a q(s',a') - q(s,a)], \tag{12}$$

where $q(s,a)$ is the of action value function. $s$ is the current state. $a$ is the current action. $s'$ is the next state. $a'$ is the action taken in the next state. $r$ is the reward obtained by the system. $\alpha$ is the learning rate. $\gamma$ is the attenuation factor. $max_a q(s',a')$ represents the next state $s'$ to select action $a'$ to maximize the value function.

Step 2. Select the next action $a$ based on the current Q value table. The Q value of the initial state is zero, and the agent can randomly choose the next action at the beginning. As the number of iterations increases, the Q value table is continuously updated, and this step will choose the action with the greatest Q value return. By transforming the Equation (12), this article can get Equation (13).

$$q(s,a) = (1 - \alpha)q(s,a) + \alpha[r + \gamma max_a q(s',a')], \tag{13}$$

According to Equation (13), Equation (14) is carried out by iteration.

$$\begin{aligned} q(s,a) &= (1 - \alpha)q(s,a) + \alpha[r + \gamma max_a q(s',a')] = \\ &(1 - \alpha)[(1 - \alpha)q(s,a) + \alpha[r + \gamma max_a q(s',a')]] + \alpha[r + \gamma max_a q(s',a')] = \\ &(1 - \alpha)^2 q(s,a) + [1 - (1 - \alpha)^2][r + \gamma max_a q(s',a')] \end{aligned} \tag{14}$$

The Equation (13) continues to be substituted into Equation (14) for iteration. By analogy, $n$ iterations are required.

Step 3. This step needs to calculate the action reward. According to the current state and reward, this article uses Bellman's equation to update the previous state $q(s,a)$, after action $a$.

$$\begin{aligned} q(s,a) &= (1 - \alpha)q(s,a) + \alpha[r + \gamma max_a q(s',a')] = \\ &(1 - \alpha)^n q(s,a) + [1 - (1 - \alpha)^n][r + \gamma max_a q(s',a')] \end{aligned} \tag{15}$$

Due to the learning rate $0 < \alpha < 1$, so $0 < (1 - \alpha) < 1$. When $n \to \infty$, $(1 - \alpha)^n \to 0$. Through the iteration of Equation (13), the final action value function is updated as Equation (16).

$$q(s,a) = r + \gamma max_a q(s',a'), \tag{16}$$

At the same time, the Q-learning algorithm may take the behavior with the highest expected reward every time. Other possible actions may not be explored during the training

process. It may even enter the "local optimum", failing to complete the path planning. Therefore, in order to avoid it, the setting function *path* is used to represent all the previous exploration points. The safety distance coefficient *D* is set to avoid entering the "local optimum". If this exploration is a point in the path, the reward *r* obtained by the system returns to 1. If the safety distance *D* < 0.5 m, the reward *r* obtained by the system returns to 0.

Step 4. Repeat step 3 until the iteration ends. The Q value table is obtained after the last iteration.

Step 5. The agent updates the action value function by interacting with the environment. The updated Q value table is used to select the optimal strategy. According to the best path selected by the Q value table, it is planned many times in the order of multi-purpose arrival.

It can be seen from Equation (16) that the Q-learning algorithm has nothing to do with the initial value. It can guarantee convergence without the environment model. This feature can be applied to robot path planning. However, according to the algorithm flow: Q-learning updates the Q value table by calculating the action value function, and then selects the optimal strategy according to the Q value table.

## 4. Multiple Destination Path Planning Simulation

### 4.1. Multi-Destination Path Planning Algorithm Simulation Steps

Based on multi-destination path planning and design the algorithm structure is shown in Figure 10.

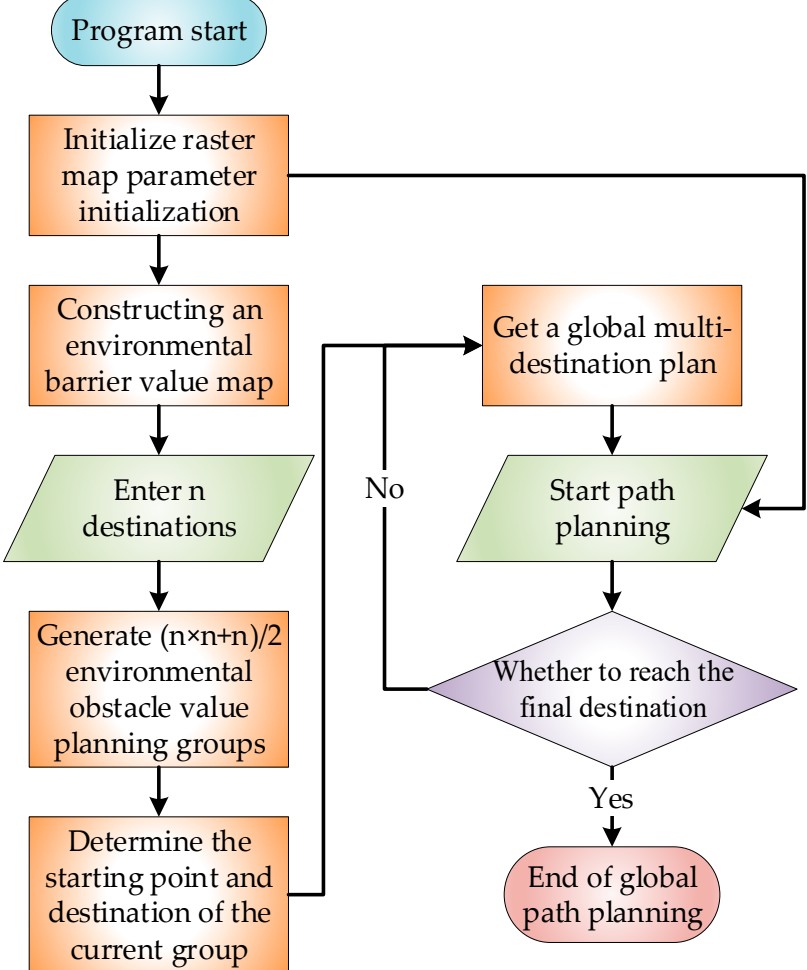

**Figure 10.** Algorithm flow of multi-destination path planning.

As shown in Figure 10, the algorithm of this article can be divided into several steps. The several concrete steps are shown in the following.

Step 1. In order to confirm $n$ destination coordinates, this article inputs the grid map model and initializes it.

Step 2. Rebuild the environmental grid map. According to the coordinates of $n$ destination, $(n^2 + n)/2$ planning matrices are generated. The planning matrices are the matrices between each two destination points. According to the planning matrices, this article calculates the environmental obstacle value.

Step 3. According to the obstacle value obtained in the previous step, this article uses the simulated annealing method to sort. The order of arrival sequence of multiple destinations is obtained.

Step 4. According to the order of multi-destination arrival sequence obtained in Step 3, this article starts executing the planning instruction. The path planning is carried out for each two target points based on the improved Q-learning algorithm in turn.

Step 5. This step needs to judge whether the current destination is the starting point. If the conditions are met, the exercise ends. Otherwise, the improved Q-learning algorithm prepares to move to the next destination point and goes to step 4.

### 4.2. Multi-Destination Simulation Experiment

In order to verify the feasibility and effectiveness of the multi-destination path planning method in this article, a multi-destination path planning simulation experiment was designed. For the convenience of comparative research, it is assumed that the grid side length of the mobile robot working environment is 1 m. On this basis, in MATLAB modeling and simulation experiment of multi-destination global path planning is carried out.

Aiming at verifying the adaptability of the multi-destination path planning algorithm to the environment, the experimental environment is set up in layers. The environment map model is $40 \times 40$. At the same time, to verify the adaptability, the algorithm can improve the transportation efficiency of mobile robots. This experimental environment has 8 sets of simulation experiments. The number of destinations in each group of experiments gradually increased. The length of the moving path obtained by the experiment is shown in Figure 11. The comparison of the planned routes for the eight destinations is shown in Figures 12 and 13. In the two figures, point **S** represents the starting point of the mobile robot. Point set $T_i$ ($i = 1, 2, 3, 4, 5, 6, 7, 8$) represents the destination of transportation. The black grid area represents 1 obstacle. The solid line represents the planning route of the mobile robot.

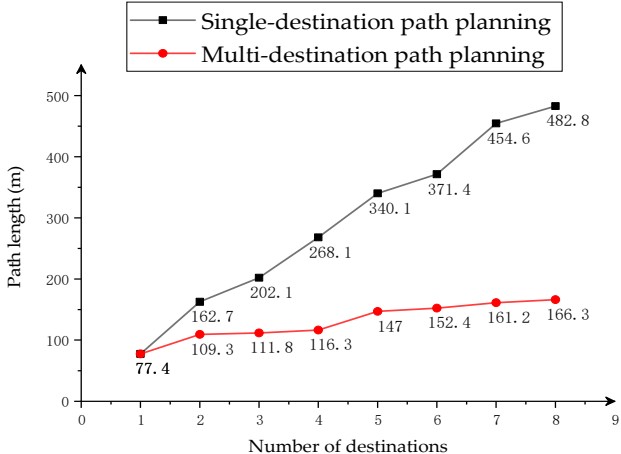

**Figure 11.** Comparison of single and multi-destination path lengths.

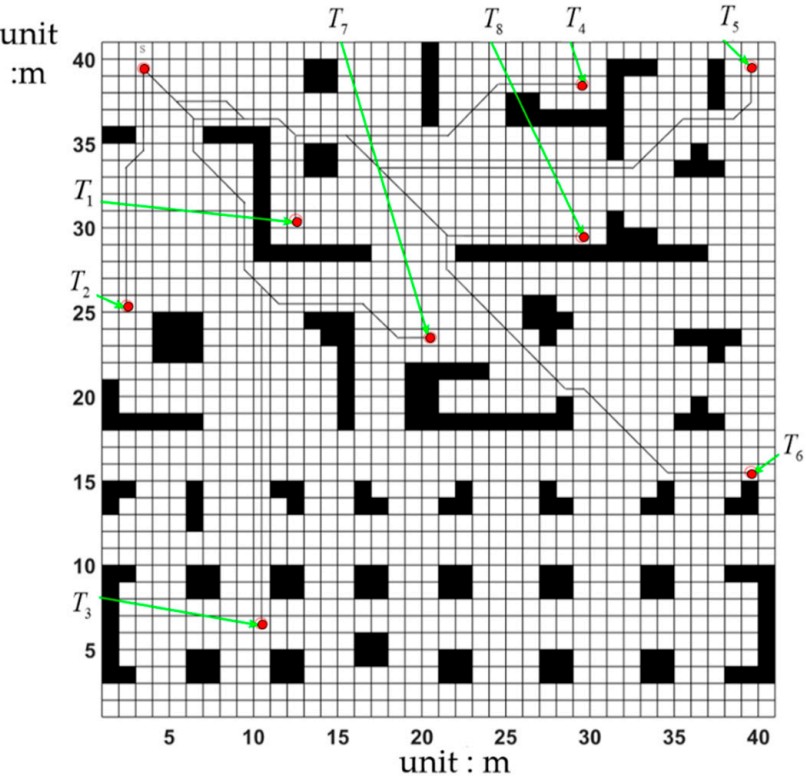

**Figure 12.** Single-destination path planning.

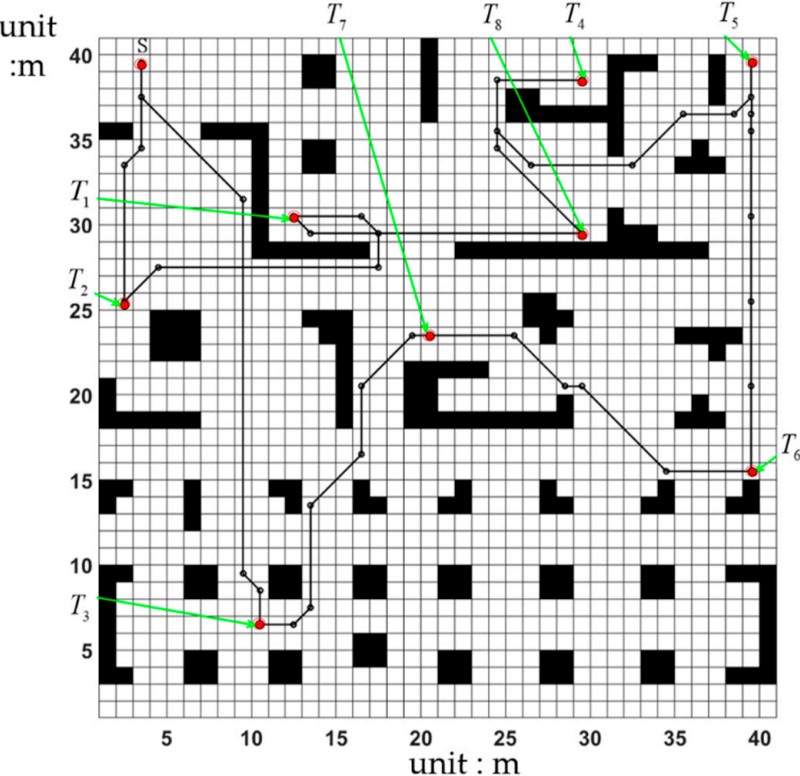

**Figure 13.** Multi-destination path planning.

It can be seen from Figures 12 and 13 that single-destination path planning causes the mobile robot to return to the starting point multiple times. It reduces the efficiency of the mobile robot. Figure 13 shows the path of multi-destination planning in this article. The

path of multi-destination planning can reach all destinations at one time. It reduces the number and time of calculation paths. From the path length comparison chart in Figure 11, as the number of destinations increases, the path length planned by the single-destination path planning algorithm increases rapidly. However, the path length planned by the multi-destination path planning algorithm designed in this article grows smoothly. Compared with the ground path planning algorithm, the average length of this article is reduced by 22%. It improves the work efficiency of the mobile robot.

### 4.3. Comparison of MD-Q-Learning (Multi-Destination-Q-Learning) Based on the Fewest Obstacles and NSGA_II

NSGA_II is one of the most popular multi-destination genetic algorithms at present. It reduces the complexity of non-inferior ranking genetic algorithms. It has the advantages of fast running speed and good solution set convergence. It has become the benchmark for the performance of other multi-destination optimization algorithms based on heuristics. MD-Q-learning algorithm based on passing through the fewest obstacles is based on heuristics. Therefore, this article compares the multi-destination method of this article with the previous NSGA_II algorithm in two test cases. This selection of test cases is relatively difficult, comparing with other previous simulation experiments. The obstacle percentages for these test cases are 18% and 34%. In order to compare these two algorithms, theoretical analysis and experimental analysis is undertaken in this article. In the theoretical analysis of these algorithms, this article estimated their time complexity.

Definition: $O(M^aN^b)$ is a function describing the space complexity of the algorithm. The smaller the values of a and b, the more concise the algorithm is. Meanwhile M is the number of iterations, N is the overall size of population.

The time complexity of algorithm in this article is $O(M^2N^2)$, when the time complexity of the NSGA_II algorithm is $O(MN^2)$. The additional complexity of multi-destination path planning based on MD-Q-learning with minimal obstacles is caused by the map reconstruction and the nature of the Q-learning algorithm. However, since the overall and iteration values are small, the running time of the algorithm will not increase. In order to prove this point, in the experimental analysis of the algorithms, their running time and the number of fitness function calls are compared.

*Path length* is the length of the planned path. *Path safe* is determined by using a potential field. The potential field of surroundings due to obstacles can be given a value by linear decreasing potential value.

This article considered the specific obstacle placement position shown in Figure 14b. This scheme is one of the complex schemes considered in earlier path planning research. This research compares the use of NSGA_II algorithm to solve the path planning problem. It compares the three representations of integer coding, binary coding and mixed coding. This research selected size of population and number of iterations of dual-target NSGA-II. Size of population is 500, and number of iterations is 800. The mutation rate (pm) was selected as 0.032, and the crossover rate (pc) was selected as 0.9. Using these values for the NSGA-II parameters, the Pareto front obtained in all three scenarios is shown in Figure 14a. Among the three schemes, it was found that the integer-encoded gene representation scheme can produce better results. The integer-encoded gene represents the minimum path distance solution. The minimum path distance solution, the minimum path vulnerability solution and the intermediate compromise solution are shown in Figure 14b.

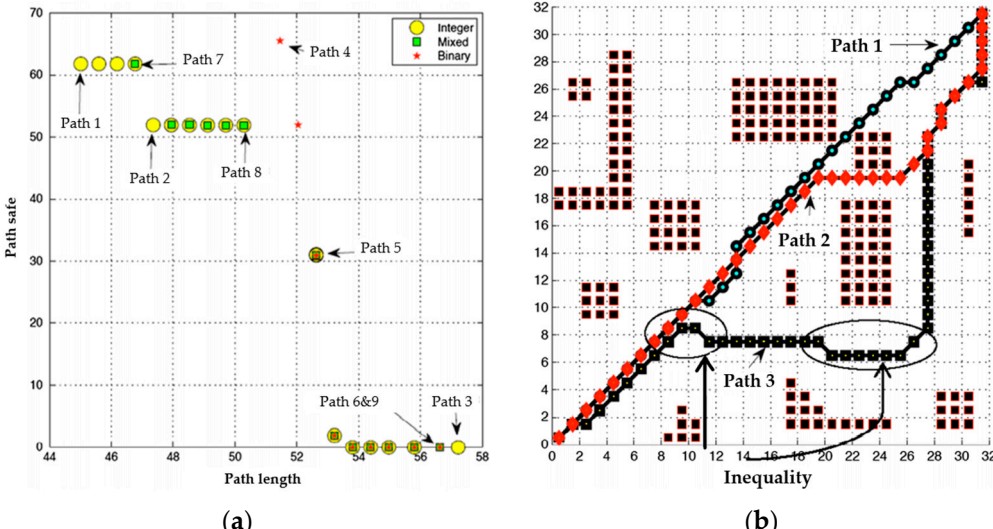

**Figure 14.** (**a**) Pareto front obtained by NSGA_II using three schemes. Integer encoding means that the scheme performs best. (**b**) Obstacles on a 32 × 32 grid use integer coding to represent the solution path obtained by the scheme.

In order to compare the performance of the proposed algorithm and NSGA_II, the previous scenario was repeated using the algorithm (based on MD-Q-learning with minimal obstacles) in this article. In this experiment, a population with a size of 50 was selected and the algorithm was run for 100 iterations. The minimum and maximum values of the local search operation are 60 and 10, respectively. Figure 15a shows the Pareto front obtained by the CSD-based MD-Q-learning algorithm. Figure 15b shows some of the obtained paths. As shown in Figure 15a, the Pareto obtained by the MD-Q-learning algorithm based on the fewest obstacles and the NSGA_II algorithm is close enough to the optimal Pareto front. The MD-Q-learning algorithm based on passing through the fewest obstacles applies more local search operations to these solutions. In this way, appropriate solutions can improve and enhance Pareto solutions.

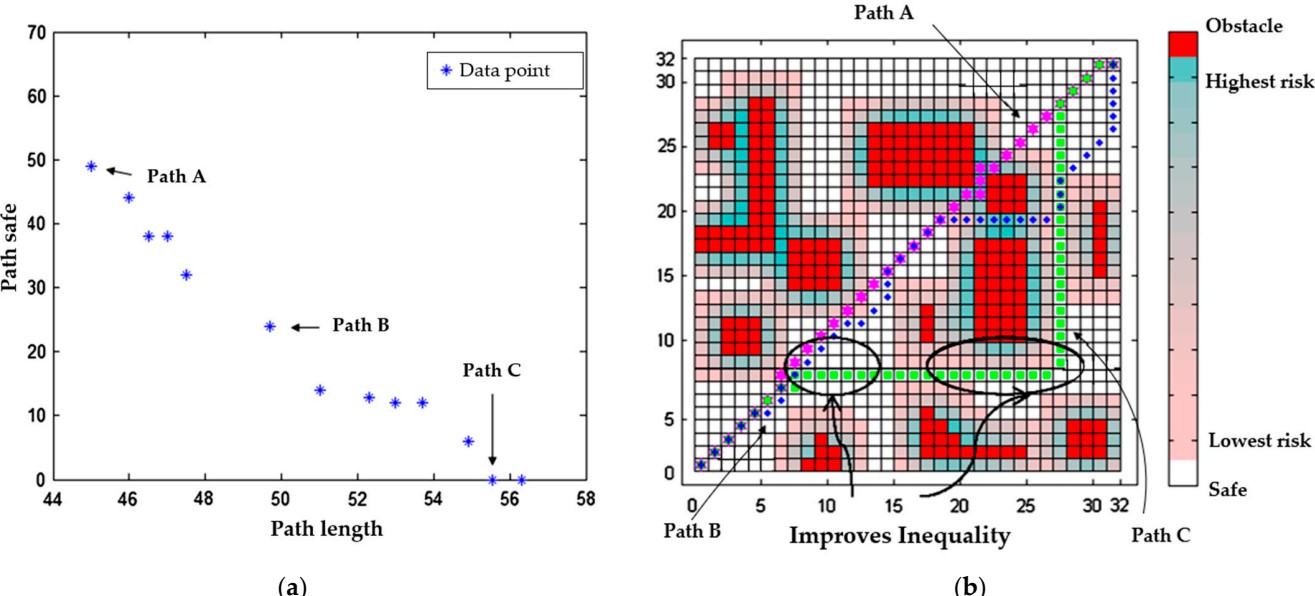

**Figure 15.** (**a**) Pareto front obtained using the MD-Q-learning algorithm based on passing through the fewest obstacles. (**b**) Use of the obstacle arrangement and three paths based on the MD-Q-learning algorithm that passes through the fewest obstacles.

In addition, the local search operator will try to improve the smoothness of the path as close as possible to the best Pareto front. Therefore, the path obtained by this article is smoother than the path shown in Figure 14b. Table 1 compares the parameter initialization and performance of NSGA_II and the MD-Q-learning algorithm based on the fewest obstacles. According to the table, the running time of the MD-Q-learning algorithm based on passing the fewest obstacles is 3.04. The running time of the NSGA_II algorithm is 2.99. The MD-Q-learning algorithm based on the fewestobstacles has fewer fitness function calls to achieve the best solution. The Pareto front obtained from the two algorithms is as close as possible to the optimal Pareto front. However, in the MD-Q-learning algorithm based on the fewest obstacles, the final coefficient of variation of Pareto front for different targets is smaller than that of the NSGA_II algorithm. Therefore, Pareto solutions obtained by the MD-Q-learning algorithm have a better distribution in the Pareto front.

As mentioned earlier, NSGA_II has $MN^2$ complexity. MD-Q-learning based on passing through the fewest obstacles has $M^2N^2$. The other complexity of MD-Q-learning based on passing the fewest obstacles is due to the nature of the proposed local operation. It has a great influence on the results of the algorithm. For better comparison, local search is merged into NSGA_II. If the proposed local search operator is added to NSGA_II, its complexity will be $O(M^2N^3)$. The complexity of the MD-Q-learning algorithm based on passing the fewest obstacles will be $O(M^2N^2)$. In addition, the first case is tested by using NSGA_II and the suggested local search. Table 1 shows the parameter initialization using local search and the performance of NSGA_II in better size of population and number of iterations. As shown in Table 2, the running time of the algorithm is 3.21 in the same size of population and number of iterations with the MD-Q-learning algorithm. In this test, the coefficient of variation of different targets is smaller than the previous NSGA_II. However, these values are much higher than the MD-Q-learning algorithm based on passing the fewest obstacles. Therefore, in this comparison, MD-Q-learning based on the fewest obstacles has better theoretical and experimental results.

**Table 1.** Comparison of parameter initialization and performance between NSGA_II and the MD-Q-learning algorithm based on the fewest obstacles.

| Algorithm | NSGA_II | CSD Based MD-Q-Learning |
| --- | --- | --- |
| Size of population | 500 | 50 |
| Number of iterations | 800 | 100 |
| Number of heuristic operations | pc = 0.9 and pm = 0.032 | $N_{LocalSearch}$ = between 10 and 60 |
| Fitness values of safest path/(length, safety) | 53.75, 0 | 55.54, 0 |
| Fitness values of shortest paths | 45, 61.25 | 45, 49.12 |
| Number of fitness function calls | 772,800 | 218,893 |
| Coefficient of variation in safety objective | 2.47 | 1.41 |
| Coefficient of variation in length objective | 2.71 | 0.437 |
| Run time of algorithm (s) | 2.99 | 3.04 |

**Table 2.** Parameter initialization and performance in NSGA_II using local search operators and the MD-Q-learning algorithm based on the fewest obstacles.

| Algorithm | NSGA_II | CSD Based MD-Q-Learning |
| --- | --- | --- |
| Size of population | 50 | 50 |
| Number of iterations | 100 | 100 |
| Number of heuristic operations | pc = 0.9 and pm = 0.032 and $N_{LocalSearch}$ = 30 | $N_{LocalSearch}$ = between 10 and 60 |
| Fitness values of safest path/(length, safety) | 53.75, 0 | 55.54, 0 |
| Fitness values of shortest paths | 45, 61.25 | 45, 49.12 |
| Number of fitness function calls | 275,645 | 218,893 |
| Coefficient of variation in safety objective | 2.03 | 1.41 |
| Coefficient of variation in length objective | 1.92 | 0.437 |
| Run time of algorithm (s) | 3.21 | 3.04 |

To test the ability of the algorithm to approach the best Pareto front, a benchmark test is added. As shown in Figure 16, there are 88 obstacle units on a 16 × 16 obstacle grid. This test was also considered in earlier path planning studies.

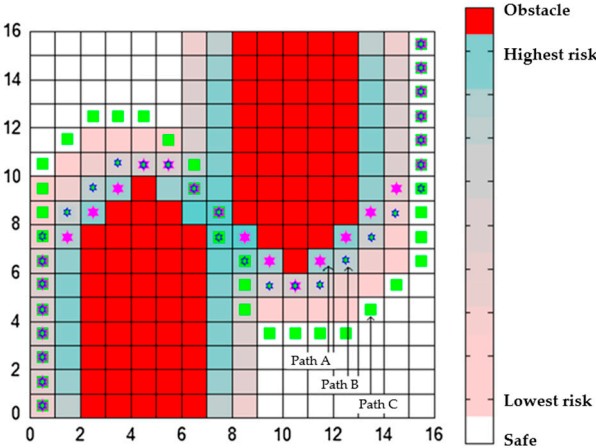

**Figure 16.** Three paths obtained under ordinary obstacle arrangement.

This experiment is performed using 50 populations and repeating 50 times. Similarly, the minimum and maximum values of the local search operation are equal to 10 and 50, respectively. As shown in Figure 16, in this case, the CSD based MD-Q-learning algorithm obtained the shortest and smoothest path (solutions A and C, respectively). Solution B shows an intermediate path. It seems that the middle path chosen makes a good compromise between the two extreme solutions. Figure 17b shows the best Pareto front of the CSD based MD-Q-learning algorithm. The obtained Pareto front shows that the minimum path length solution and several other solutions are superior to the earlier reported solutions. The algorithm in this paper is heuristic, so the failed Pareto solution will be obtained (represented by "path X" in Figure 17b). The main reason is the limitation of minimum path length and some safety factors.

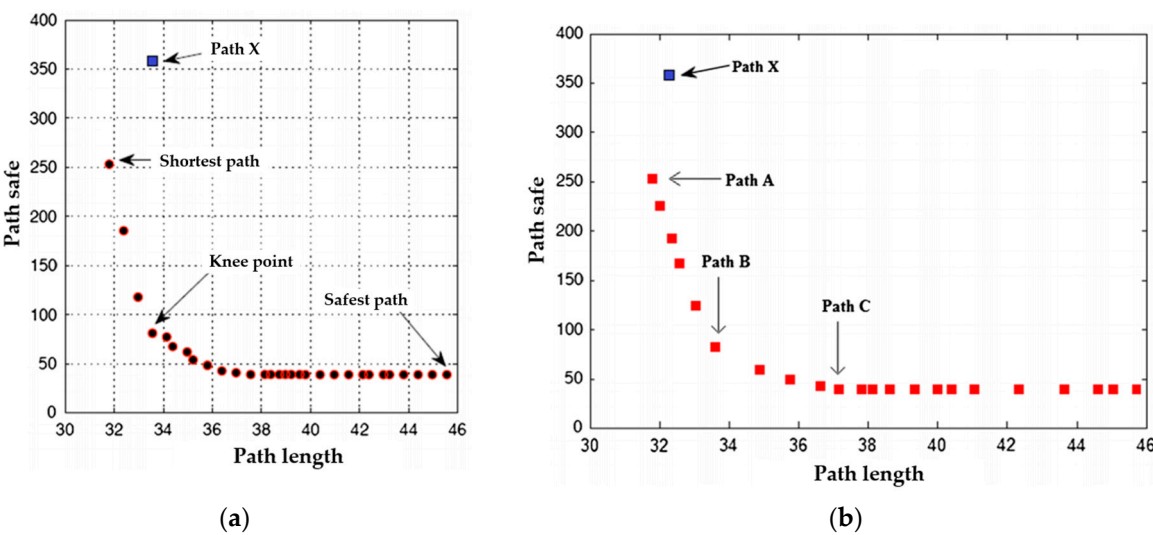

**Figure 17.** (**a**) In the fourth case test, the Pareto front is obtained using the three presentation schemes in NSGA II. (**b**) In the fourth case test, the Pareto front obtained by the CSD based MD-Q-learning algorithm was used.

Moreover, compared with the research results shown in Figure 17b and the research results of NSGA II, it has a better distribution throughout the Pareto front. Therefore, it

shows the superiority of our proposed program and its ability to achieve close to the best Pareto front.

## 5. Conclusions

Aiming at the multi-destination path planning scenario, this article designed a multi-destination path planning algorithm. The algorithm combines the sorting method based on the reconstructed environment grid map and the improved Q-learning algorithm. Firstly, it is necessary to calculate the environmental obstacle values between the starting location and the destination. At same time, the environmental obstacle values need to be calculated between the two destinations multiple times. Then, according to the calculated environmental obstacle value between the two locations, multiple destinations are optimized and sorted. The global destination path sequence with the smallest environmental obstacle value is selected. Finally, based on CSD's MD-Q-learning, multi-destination global path plan is obtained.

The evaluation function based on CSD increases the possibility of the MD-Q-learning algorithm to reach the optimal Pareto front. It is used to solve the robot path planning problem. Using the proposed multi-destination path planning method, the Q-learning algorithm is extended to solve the multi-destination problem. The implementation results show that the Pareto front of the proposed multi-destination path planning method is close to the optimal Pareto front. In addition, the search operation is distributed throughout the Pareto front based on the CSD evaluation function and the sorting method with passing through the fewest obstacles. Therefore, it is not only close to the best Pareto front, but also passes through fewer obstacles. The comparison results of the MD-Q-learning and NSGA_II algorithms based on the fewest obstacles show that both algorithms are close to the best Pareto front at an acceptable level.

The simulation results show that the multi-destination path planning method designed in this article has good adaptability in different environments. The length of path in the global planning is reduced by 22% compared with the average length of the single-destination path planning algorithm. The work efficiency of the mobile robot is improved. Comparing with other multi-destination path planning algorithms, NSGA II, the solutions obtained by the algorithm of this article are close to the best Pareto front at an acceptable level. Moreover, the mobile robot safely passes fewer obstacles.

This article did not consider the local path planning. There is no route smoothness optimization. The detail optimization of the path is insufficient. Therefore, there is still a large space to reduce the length of path. At the same time, the greedy algorithm leads to a lot of redundant searching. These redundant searches lower the calculation speed. The algorithm in this article and NSGA_II belong to the heuristic algorithm. In some special cases, the heuristic algorithm will get very bad answers or poor efficiency. In the future, this algorithm needs to be further improved and applied to real robots.

**Author Contributions:** Conceptualization, H.Z. and K.D.; methodology, H.Z. and K.D.; formal analysis, Y.Q. and N.W.; data curation, H.Z. and K.D.; writing—original draft preparation, H.Z. and K.D.; writing—review and editing, H.Z.; visualization, K.D. and N.W.; supervision, H.Z. and L.D.; funding acquisition, H.Z. All authors have read and agreed to the published version of the manuscript.

**Funding:** This research was funded by the National Natural Science Foundation of China, grant number 51505335; the Doctor Startup Projects of TUTE, grant number KYQD 1806.

**Institutional Review Board Statement:** Not applicable.

**Informed Consent Statement:** Not applicable.

**Data Availability Statement:** Not applicable.

**Conflicts of Interest:** The authors declare no conflict of interest.

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
