# Peer review of "Multi-Destination Path Planning Method Research of Mobile Robots Based on Goal of Passing through the Fewest Obstacles"

_applsci, doi:10.3390/app11167378_

Round 1

Reviewer 1 Report

The paper presents an algorithm for path planning for mobile robots, based on a raster-representation. Path planning is done for multiple destinations. Further, the path is optimised regarding the number of obstacles passed. The work looks interesting, and the results promise good practical results in comparison to the NSGA II algorithm.

However, the algorithm is difficult to understand. There is a variety of sentences and formulas that I do not understand. Thus, it is difficult to judge what the contribution of the paper is. The results indicate that the algorithm can be applied to practical applications.

There are language issues. The sentences as such are grammar-wise correct, but their composition does not make sense on a variety of occasions.

The algorithm claims to provide multi-destination path planning. However, it is unclear how this is achieved. For example, are several solutions of single-destination planning combinatorically combined? Or is there a different mechanism? This remains unclear but has an impact on the complexity of the algorithm. This seems to be described ca. line 64, but remains unclear. For example, you mention "between two points", but refer to multi-destination global path planning. Further, line 65: "censured" is probably not the correct word.

line 89-90: incomprehensible.

line 112: incomprehensible. What is "two aspects Priority"?

line 113: incomprehensible "convenience of elaboration"?

line 116/117: incomprehensible and grammar wrong. What is the "optimal path with high possibility"?

line 126: Picture -> Figure ?  "chose" -> "chosen"

The environmental grid map is not explained sufficiently. I understand from the example later in the article that you transform the obstacle map using a 3x3 filter. This algorithm seems to use heuristics, that are not further discussed. The role of the environmental grid map is not entirely evident.

line 133: incomprehensible: there are many situations in the movement sequence.

line 134: passes --> pass

Inconsistent use of the variables n and m, later N and M. Please define what these represent.

line 150: incomprehensible: "displayed shelves"

line 150: vague: "certain generality"

lines 157ff: incomprehensible. What are the grid and adjacent grids? what are environmental obstacle values?

Equation 1: incomprehensible. note also that there is an index error, e.g., for i = 0 or j = 0 and i = 40 or j = 40.

The following description and formulas are rather incomprehensible.

Equation 3 is inconsistent.

line 188: Why "This article" ?. check grammar.

line 196: what is the "rule of Metroplis". Do you mean "Metropolis"? But this does not become clearer, unfortunately.

equation 8: incomprehensible

line 209: incomprehensible. "In this section, this article ..." ???

line 227: paragraph is incomprehensible. It is unclear how Q-learning is used?

Figure 7: algorithm and its variables are incomprehensible

line 245: incomprehensible

line 246: incomplete sentence

line 263: The Q-learning algorithm ...

line 283: what is the number of states? unclear. Incomprehensible: "the basic form of the algorithm is Equation 9"

line 294: is this an iteration or a recurrence?

line 300: unclear. you write a element (0,1). What does this mean? When 0<1-a < 1, then also 0<a<1 is valid. This is incomprehensible.

line 333: what is n here?

line 334: what are the "planning matrices"?

line 342: incomplete sentence: "The condition needs to determine".

What is the complexity of the algorithm? It also seems to have a very high memory complexity.

line 386: what does M2N2 mean?. also: MN2?

line 400: why 500 people? do you mean instances? iterations? entire paragraph is difficult to understand.

line 420: incomprehensible

line 433: incomprehensible

line 435/436: incomprehensible

Table 1: Why do you use different values for the size of population and number of iterations for the two algorithms? For the sake of comparison, these should be the same.

line 493: you write multi-destination, but the main algorithm does only handle single destination.

line 494: vague and incomprehensible: "It has a certain generation ability"

line 497: incomprehensible: what is the "stability of the robot"?

It is difficult to verify the conclusion as large parts of the article are incomprehensible. The algorithm seems to be based on heuristics. Their validity is not further discussed in the paper.

Reviewer 2 Report

The paper deals with path planning method research for mobile robot navigation. The topic is actual and interesting. The problem is solved using the simulations by Matlab software.

Introduction is sufficient for this topic. Authors sumarize the actual state of the art.

The proposed methods are adequate to solved problematic. Multi-destination path plannig schemes were designed. Also the environmental grid map techniques are used for path planning.

Multi destination global path planning strategy based on improved Q-learning algorithm is presented. The improved Q-learning algorithm i sused to complete global path planning. Detailed description is also mentioned.

Multiple destination path planning simulation is also presented in the paper. The algorithmus is clearly described and presented. Simulation results are correctly presented and discussed. The results shows advantage of multiple destination path planning in compare with single destination path planning method. The benefits and contribution from using of the multiple destination path planning is clear.

Comments for improvement:

It is clear that research on this issue is not complete. Therefore, such a scientific article should also include plans for the future in this research. The presented simulations show the viability of the presented methodology, but for implementation in real robots it is necessary to perform other activities. Please discuss this issue and activities in more detail.

Reviewer 3 Report

  1. In summary, the authors state that the method is implemented in Matlab, but from the paper it appears that it is a modeling and a simulation in Matlab. Please clarify.
  2. At related work it is recommended to consult the reports: - Novel Bioinspired Approach Based on Chaotic Dynamics for Robot Patrolling Missions with Adversaries, Entropy, vol. 20, no. 5 pp 378, 2018. - Imparting protean behavior to mobile robots accomplishing patrolling tasks in the presence of adversaries, Bioinspiration & Biometics, Vol: 10, Issue: 5, Pages: 56017-56017, Oct. 2015
  3. It is recommended to insert in the text fig. 1 after referring to it.

4. It is recommended to reread the manuscript and correct the expression in English.

5.a) The authors refer in the description of the method, in expression, to the rules of the game of go and the chess. Not all specialists in the field know these rules. It is recommended either to give up this expression or a more detailed description of the rules in correlation with the mathematical description of the method.

b) But the article does not present a mathematical description of the algorithm. It is recommended to introduce the mathematical description of the algorithm through clear relations, implementable in calculations.

c) It is recommended the mathematical presentation of the corresponding algorithm Fig. 4, so that the readers can reproduce the method and calculations.

6. It is recommended to reread the article and correct the editing mistakes.

7. In 2.2 in the first sentence line 173 the authors use the concept of 'strong robustness'. Specify what it is referring to. Define it, give examples, and in the discussions exemplify the way in which the work ensures the achievement of this robustness and how "strong" is it.

8. Explain how the following variables are determined: "The initial temperature ????? , the end temperature ????? and the cooling rate ? (0<r<1)". (line 177)

9. It is recommended to reread the article and define all the notions that are not defined, which are stated without giving details how they were calculated.

10. Line 209 states that the paper try to obtain the 'optimal solution'. Explain what the 'optimal' consists of, how you define it, what values ​​are actually obtained, present a broad discussion on this 'optimal' in a final discussion section.

11. Reread the paper and give explanations for each term introduced but which is not explained, is not defined mathematically and is not discussed in the paper. Eliminate terms, concepts, attributes, adjectives, attributes that you do not define mathematically and do not discuss in the end.

12. Give more detailed mathematical relations for the algorithm from 3.1 Q-learning algorithm optimization.

13. The same request for algorithm from Fig. 9.

14. The same request for algorithm from Fig. 10.

15. As a result of the above observations it is necessary to develop a mathematical description of the methods.

16. A large section of discussion is necessary to make analysis on the results and methods.

Reviewer 4 Report

Dear Authors!

With great interest, I’ve got acquainted with your paper. It is devoted to a rather urgent problem, namely, the multi-destination path planning for mobile robots. The domain rapidly is developed in recent years making your article a fine contribution to the field. The paper presents an algorithm based on the reconstructed environment grid map, which is a quite common approach in path planning algorithms. The contribution of the research consists in the application and improvement of Q-learning algorithm to solve the task that guarantees convergence. Also, the list of references contains recent works, a number of them were written in the last 3 years.

Nevertheless, while reading the paper some questions and remarks arise.

First, is ‘CSD-based MD-Q-learning algorithm’ the algorithm you’ve developed? If yes, then clarify it. If no, I cannot find the comparison of your algorithm with other multiple path planning algorithms.

The manuscript is not well-designed. Some phrases are not finished by a dot when other phrases start. Captions are not well highlighted. Figures require clarification.

Please, improve the quality of English and of illustrations. E.g., in Fig. 8, it is better to make red circles (destination points) more visible. The same is for Fig. 12 and 13.

A minor list of text flows is given below.

Line 18: please clarify what is ‘q value of the path planning’ in the Abstract. The same is for ‘a’ and ‘b’ in Line 21.

In Line 271 and 272, you use two different ways to spell ‘Q value’ and ‘Q-value’. Please choose the uniform spelling.

In table 1, in ‘Fitness values of shortest0 paths’, the symbol ‘0’ seems to be redundant.

Round 2

Reviewer 1 Report

Thank you for resubmitting your paper. The authors have made minor changes in the manuscript. Unfortunately, major comments from the reviewer were not sufficiently addressed or have been misunderstood.

In your revised manuscript, no line numbers were present. Thus, I describe the issues by other means.

In Line 12 of the abstract: incomprehensible. "The Pareto numerical graph is gotten" and also the following sentence. I don't understand what you mean; sorry.

Section 2, first paragraph, second last line: What does the yellow N mean in this context?

Figure 3(b): note that the colour scale is discrete (i.e., an integer value).

Page 4, last paragraph: what are 2.1 and 2.2? Are these sections?

Following Eq. 1, the Me matrix will have a line with the value 0 on the upper row and the column most to the right. I assume that you start with the index 1 and go up to 40. That could mean that the algorithm could fail at the borders.

Figure 5: Please also provide the Me Matrix in this figure.

Eq. 2 is still incomprehensible. It seems that you calculate a weighted sum of the sub-matrix. The "x" should be a normal multiplication.

Step 4: "..." is duplicated. Please remove one of these.

Equation 4: There is no obvious rule for the indices in the matrix. Why is the second line on the bottom? Should the second column be: "T_1, T_2, ..., T_n"? Then the third column would be "T_2, T_3, ..., T_n, T_1". The second-last row would then be "T_n, T_1, ... T_n-2, T_n-1".  If I did misunderstand this construction rule, please explain how the matrix is constructed.

I don't understand most of Section 3, sorry.

Page 9: 0.707: Is this 1/sqrt(2)? Further, something is wrong with Eq 11. Please check carefully the ranges of all rows. Further, are the indexes correct? An index of 0 (which is possible according to Eq 11) is out of range (cf. your definition that the matrix is 40x40 with ranges from 1 to 40). So, either the index 0 or the index 40 are invalid, depending on if you count from 1 or from 0.

Page 10: is "constantly updated": do you mean "continuously updated"? Further, the following sentence is incomprehensible. ("... article can get equation ..." does not have a meaning).

After Eq13: the rate alpha cannot be an element of something, as alpha is not a discrete value. alpha can be in the range, which is usually expressed with 0<alpha<1.

Safety distance D=0.5m: This implies that your grid is a metric grid. This is not explained earlier.

The algorithm in Section 4.1 has a very high spatial complexity. This must be considered.

Figure 13: The outcome is not optimal. One could for example draw a graph from S to T2 to T3 .... to T8 to T1 to S.

Also providing the Me matrix for this example would help to understand your algorithm.

Page 13: the overall size of what?

Page 14: you still write 500 people. This means "500 persons". However, this does not make sense in this case. It is further unclear what the population consists of. Please revise.

For the claims regarding Pareto, no proof is given.

Tables 1 and 2: Please provide a table where you compare both algorithms under comparable conditions (e.g., the same size of the population, the same number of iterations, etc.). Please put these in the same table, so that things can be compared.

Second-last paragraph: incomprehensible. What do you mean by "is closed to"?

Further: "pass" -> "passes".

Please identify the cases where the heuristics might fail. Knowing this is important to select appropriate application areas for your algorithm.

Reviewer 3 Report

The authors did not make the recommended changes, but the paper has a degree of sufficiency that allows publication.

Author Response

Dear Reviewer 3,

Manuscript ID: applsci-1311309

Our deepest gratitude goes to the anonymous reviewer for your careful work and thoughtful suggestions that have helped improve our manuscript substantially.

Meanwhile, we would like to thank you for your affirmation of our manuscript.

According to your valuable suggestions, this manuscript becomes more perfect for readers.

Thanks for all the help.

Best wishes,

Dr Hongchao Zhuang

Reviewer 4 Report

Dear Authors!

Thank you for the mistakes and flaws correction. The manuscript has become acceptable.

Author Response

Dear Reviewer 4,

Manuscript ID: applsci-1311309

Our deepest gratitude goes to the anonymous reviewer for your careful work and thoughtful suggestions that have helped improve our manuscript substantially.

Meanwhile, we would like to thank you for your affirmation of our manuscript.

According to your valuable suggestions, this manuscript becomes more perfect for readers.

Thanks for all the help.

Best wishes,

Dr Hongchao Zhuang